# Effective Absorption of Dichloromethane Using Carboxyl-Functionalized Ionic Liquids

**DOI:** 10.3390/ijerph20105787

**Published:** 2023-05-11

**Authors:** Mengjun Wang, Manman Zhang, Shaojuan Zeng, Yi Nie, Tao Li, Baozeng Ren, Yinge Bai, Xiangping Zhang

**Affiliations:** 1College of Chemical and Engineering, Zhengzhou University, Zhengzhou 450001, China; 2Longzihu New Energy Laboratory, Zhengzhou Institute of Emerging Industrial Technology, Zhengzhou 450000, China; 3CAS Key Laboratory of Green Process and Engineering, State Key Laboratory of Multiphase Complex Systems, Beijing Key Laboratory of Ionic Liquids Clean Process, Institute of Process Engineering, Chinese Academy of Sciences, Beijing 100190, China; 4Langfang Green Industrial Technology Center, Langfang 065000, China; 5College of Chemical Engineering and Environment, China University of Petroleum, Beijing 102249, China

**Keywords:** ionic liquids, dichloromethane, absorption, vapor-liquid equilibrium, NRTL model, quantum chemical calculations

## Abstract

Dichloromethane (DCM) is recognized as a very harmful air pollutant because of its strong volatility and difficulty to degrade. Ionic liquids (ILs) are considered as potential solvents for absorbing DCM, while it is still a challenge to develop ILs with high absorption performances. In this study, four carboxyl-functionalized ILs—trioctylmethylammonium acetate [N_1888_][Ac], trioctylmethylammonium formate [N_1888_][FA], trioctylmethylammonium glycinate [N_1888_][Gly], and trihexyl(tetradecyl)phosphonium glycinate [P_66614_][Gly]—were synthesized for DCM capture. The absorption capacity follows the order of [P_66614_][Gly] > [N_1888_][Gly] > [N_1888_][FA] > [N_1888_][Ac], and [P_66614_][Gly] showed the best absorption capacity, 130 mg DCM/g IL at 313.15 K and a DCM concentration of 6.1%, which was two times higher than the reported ILs [Beim][EtSO_4_] and [Emim][Ac]. Moreover, the vapor–liquid equilibrium (VLE) of the DCM + IL binary system was experimentally measured. The NRTL (non-random two-liquid) model was developed to predict the VLE data, and a relative root mean square deviation (rRMSD) of 0.8467 was obtained. The absorption mechanism was explored via FT-IR spectra, ^1^H-NMR, and quantum chemistry calculations. It showed a nonpolar affinity between the cation and the DCM, while the interaction between the anion and the DCM was a hydrogen bond. Based on the results of the study of the interaction energy, it was found that the hydrogen bond between the anion and the DCM had the greatest influence on the absorption process.

## 1. Introduction

Dichloromethane (DCM) is an excellent organic solvent [1,2] which is widely used in film, metal manufacturing, and pharmaceutical production fields. The annual world production of DCM exceeds 5 × 10^5^ tons [3], and due to its highly volatile nature, approximately 77% of the emitted DCM is released into the atmosphere [4]. The release of DCM into the atmosphere affects the environment by producing highly toxic phosgene and carbon monoxide and destroying the ozone layer [5]. DCM also poses a serious threat to human health [6] particularly as risk of cancer [7]. Therefore, it is urgent to develop DCM recovery technology.

To date, various recovery methods for DCM have been developed, such as the adsorption [8,9,10,11] and absorption methods [12,13,14,15]. Nevertheless, it cannot be ignored that the adsorption method has the problems of low separation selectivity and limited adsorption capacity of the adsorbent [16]. The absorption method is considered a promising DCM capture technology due to its good separation selectivity, the applicability of the gas source, and its simple process. At present, silicone oil and several organic solvents, such as bis(2-ethylhexy) phthalate (DEHP), bis(2-ethylhexy) adipate (DEHA) [17], and polydimethylsiloxane (PDMS) [18], are used to capture DCM. However, these absorbents showed low capacity for soaking up DCM compared with other absorbents, which makes high solvent consumption achieve deep removal. Moreover, organic solvents have volatile properties, resulting in resource waste. It is of great importance to search for and develop more efficient, greener, and less volatile absorbents to overcome the above problems.

ILs have various favorable characteristics, such as low vapor pressure [19,20,21] and designable structure, and have attracted much attention in various fields. For the past few years, many studies have been carried out on the absorption of DCM by ILs, and some progress has been made. Castillo et al. [22] determined the Henry’s law constants (*K*_H_) of DCM in 23 kinds of hydrophobic ILs at 298 K and infinite dilution by the static headspace method, and found that the *K*_H_ of DCM in ILs was 14–50 times lower than that in water. Wu et al. [23] synthesized six ILs with imidazolium-based cation and one IL with pyridyl-based cations as absorbents to capture DCM. Among them, [Bmim][SCN] had an excellent absorption capacity of 1.46 g/g at 303.15 K and DCM concentration of 60% (volume fraction). According to Shi et al. [24], the most remarkable DCM absorption was obtained by using [Beim][EtSO_4_] with a capacity 0.426 g/g under an absorption temperature of 313.15 K and a DCM concentration of 37.7%. Gui et al. [14] investigated the removal ratio of DCM by using [Emim][Ac], and the results demonstrated that the removal ratio could reach 91.82% under 20 °C and ambient pressure with a gas flow rate and IL flow rate of 1 L/min and 20 mL/min, respectively. It was also pointed out that the strong interaction between anion [Ac]^−^ and DCM plays the main role in the separation ability of ILs. In the study of DCM absorption with ILs, Wu et al. [25] established a framework for multiscale research based on the COSMO-RS model. Among the screened ILs, [EMPIP][Ac] was found to be the most effective IL for absorption of DCM, and the *K*_H_ was 3.29 kPa. They also proposed that noncyclic cations, such as quaternary amine and quaternary phosphonium bases, were more conducive to DCM absorption. Moreover, the *K*_H_ values of other kinds of anions were relatively higher than those of carboxylate-based anions.

It can be seen from the above research that ILs have good performance in the capture of DCM. However, the absorption capacity of DCM is still not very high and needs to be further improved. At present, most reported ILs are conventional ILs with short-chain cyclic cations, such as imidazolium-based cations. In fact, noncyclic cations, such as quaternary ammonium and quaternary phosphonium and carboxylate-based anions have low *K*_H_ values, according to the simulation results from Wu et al. [25]. Therefore, four carboxyl-functionalized ILs were synthesized in this study: [N_1888_][Ac] (trioctylmethylammonium acetate), [N_1888_][FA] (trioctylmethylammonium formate), [N_1888_][Gly] (trioctylmethylammonium glycinate), and [P_66614_][Gly] (trihexyl(tetradecyl)phosphonium glycinate), respectively. Several factors, such as IL structure and absorption temperature, were studied to evaluate their impact on absorption performance. Then the vapor–liquid equilibrium (VLE) experiment of DCM + IL systems was carried out, and a comparison of the experimental data and the data predicted by the NRTL model was conducted to verify whether this model was suitable for the system in this study. Finally, spectroscopy analysis and quantum chemical calculations were performed to reveal the absorption mechanism.

## 2. Materials and Methods

### 2.1. Materials

Nitrogen (N_2_, >99.99%) was purchased from Henan Yuanzheng Special Gas Co., Ltd., (Henan, China). Trioctylmethylammonium chloride ([N_1888_][Cl], 97%) was purchased from Shanghai Maclin Biochemical Technology Co., Ltd., (Shanghai, China). Trihexyl(tetradecyl)phosphonium chloride ([P_66614_][Cl], 97%) was purchased from Jiangsu Aikang Biomedical Research and Development Co., Ltd., (Jiangsu, China). Anhydrous ethanol (≥99.7%) was obtained from Wuxi Yatai United Chemical Co., Ltd., (Wuxi, China). The following chemicals were also used in this study: DCM (≥99.5%), sodium acetate (99%), sodium formate (≥98.5%), glycine (99%), and Ambersep 900(OH) ion exchange resin, which were all purchased from Shanghai Titan Technology Co., Ltd., (Shanghai, China). All the chemicals used in this study are listed in Appendix A.

### 2.2. Synthesis of ILs

[N_1888_][Gly] and [P_66614_][Gly] were prepared via the anion exchange method, as reported in the literature [26,27]. Firstly, a solution of [N_1888_]Cl in ethanol flowed through a chromatography column containing anion exchange resin to obtain the [N_1888_]OH-ethanol solution. The flow rate was controlled to ensure the complete exchange of chloride. Then 1.1 mol equivalents glycine was added to the solution flowing down the chromatography column. The mixed solution was stirred with using a magnetic heating agitator for 24 h at a temperature of 313.15 K. Then the ethanol was moved at 333.15 K under vacuum and acetonitrile was used to remove unreacted glycine. Finally, the obtained light yellow viscous liquid was dried for 48 h in vacuum at 343.15 K. [P_66614_][Gly] was synthesized using the same method.

[N_1888_][Ac] and [N_1888_][FA] can also be synthesized by using the above method. However, since the anion exchange method was complicated and difficult to use on a large scale, the two ILs were prepared using the ion exchange method. This synthesis route was relatively simple, and the generated byproduct NaCl can be easily removed during preparation [28]. The detailed synthesis method was as follows: firstly, sodium acetate (0.077 mol) was added to the ethanol solution of [N_1888_][Cl] (0.07 mol). The reaction was stirred for 24 h at 323.15 K in water. Afterward, a sand core funnel was used to remove the byproduct NaCl. The ethanol was removed at 333.15 K. Finally, IL was dried for 48 h at 343.15 K. A pale yellow viscous liquid [N_1888_][Ac] was obtained. In addition, [N_1888_][FA] was synthesized similarly. The structure of these four ILs was shown in Figure 1, and the general preparation route can be found in Appendix A.

### 2.3. Characterization of ILs

The ^1^H-NMR spectra of the ILs were measured by using a Bruker 600 spectrometer with deuterated dimethyl sulfoxide (DMSO˗d_6_) as the solvent. FT-IR spectra in the range of 500~4000 cm^−1^ were obtained by using a Fourier transform infrared spectrometer (Thermo Fisher Scientific, MA, USA). The thermal decomposition of the ILs was determined by using TGA Q5000 V3.15 with a heat rate of 10 K/min under nitrogen atmosphere.

The density of the ILs was measured using an Anton Paar DMA 5000 M densitometer with a standard uncertainty of 0.000007 g∙cm^−3^, and the Anton Paar Lovis 2000 ME micro viscometer was used to measure the viscosity of ILs with an uncertainty of 0.5%. The experiments were conducted at atmospheric pressure and temperature from 313.15 K to 353.15 K with intervals of 10 K. The mass fraction of the water in the ILs was determined using a Karl Fischer moisture meter (Model C20s, Mettler Toledo Switzerland, Greifensee, Switzerland), and the water contents of these four ILs were lower than 2000 ppm. The detailed information can be found in Appendix A.

### 2.4. Apparatus and Procedure of Gas Absorption

The absorption experiment was performed using the bubble method. During the experiment, as carrier gas, nitrogen flowed into a three-necked flask containing DCM. The temperature of the condensing tube was adjusted by a cryostat cooler, and stable DCM vapor with different concentrations was obtained by adjusting the temperature of the condensing tube. The DCM gas concentration was analyzed using a gas chromatograph (GC-7960 plus, Tengzhou Allen Analytical Instrument Co., Ltd., Zaozhuang, China) equipped with a packed column (type: DNP, size 3 m × 3 mm). At the beginning of each experiment, the IL sample (about 5.0 g) was weighed using an electronic analytical balance (Type PL403, Mettler Toledo Switzerland) with a resolution of 0.0001 g [29] and placed in an absorption bottle. The absorption bottle was put in a large beaker of water, and the water level was always kept higher than the IL level in the absorption bottle. The temperature was controlled by a magnetic heating stirrer. The exhaust gas passing through the IL entered an ethanol absorption bottle and absorbed the remaining DCM. When the concentration of the DCM gas in the inlet and outlet of the absorption bottle was nearly identical, the experiment was considered to have reached absorption equilibrium. The saturated absorption capacity (*S_AC_*, unit mg/g) of the IL at different DCM concentrations and absorption temperatures was measured experimentally with an electronic balance with 0.0001 g precision. In addition, *S_AC_* can be calculated from Equation (1):(1)$$S_{AC}=\frac{m_{3}-m_{2}}{m_{2}-m_{1}}\times 1000$$
where *m*_1_ represents the mass of the empty absorption bottle, *m*_2_ represents the total mass of the absorption bottle and IL before absorption, and *m*_3_ represents the total mass of the absorption bottle and of the IL after absorption. Each experiment was carried out three times, and the average value of the three experiments was taken as the final experimental result. The experimental procedure is shown in Appendix A.

### 2.5. VLE Experiments

The vapor pressures of the DCM + IL binary system at different DCM concentrations and temperatures were measured by a VLE experimental device (Type DPCY-6C, Jiangsu Nanjing Nanda Wanhe Technology Co., Ltd., Nanjing, China), as shown in Appendix A.

Before the experiment, the IL were poured into a 500 mL beaker and then dried under vacuum for 12 h under the set conditions. Later, the pretreated IL was sealed for use. When performing this experiment, the airtightness of the experimental device should first be examined, and then the condensing device should be turned on. In our study, approximately 18 mL of the mixed solution of DCM + IL binary system was added to the U-shaped tube. Then we turned on the vacuum pump to vacuum the system, discharging the air between the liquid in the glass ball and the liquid in the U-shaped tube. After pumping air for a while, the liquid in the U-balance tube bulged out upward in a bubble state. When the bubbling had lasted for several minutes, we closed the atmospheric valve and connected the vacuum pump valve, and the air was considered clean. Then we opened the valve of the atmosphere slowly and let a small amount of air into the system, until the liquid level of both arms of the U-shaped tube was equal and maintained for about 20 min, which can be considered to reach VLE at this time. Finally, the value of the pressure indicator, atmospheric pressure, and temperature of the thermostatic water bath was recorded. The temperature reading of the experimental apparatus was accurate to 0.01 K, and the pressure was accurate to 0.01 kPa. Two repetitions of each experiment were performed, and the average value of the two experiments was taken as the final experimental result.

### 2.6. Computational Methods

Gaussian 09, revision D.01 [30] was used for quantum chemistry calculations on the basis of density functional theory (DFT). First, Gaussview was used to generate the initial structure of the DCM as well as the anion and cation of ILs, then the geometric optimization of DCM, ILs, and their complex was carried out under the B3LYP/6-311+G(d, p) [31,32] basis set. To improve the accuracy of dispersion effects [33], Grimme’s DFT-D3 dispersion correction was applied. The potential energy of the structures was at a minimum after optimizing, and the frequency check did not identify an imaginary frequency. Finally, the interaction energy of IL with DCM was calculated using Equation (2) [34]:(2)$$\Delta E\left(\mathrm{kJ}\cdot {\mathrm{mol}}^{-1}\right)=E_{\mathrm{IL}-\mathrm{DCM}}-E_{\mathrm{IL}}-E_{\mathrm{DCM}}+E_{\mathrm{BSSE}}$$
where E_IL−DCM_ signifies the energy of complex of IL and DCM, E_IL_ and E_DCM_ denote the energy of IL and DCM, respectively, and E_BSSE_ stands for the energy correct of basis set superposition error (BSSE) by the counterpoise method [35].

## 3. Results and Discussion

### 3.1. DCM Capture with Carboxyl-Functionalized ILs

#### 3.1.1. Effect of IL Structure on Absorption Capacity

The saturated absorption capacity in four carboxyl-functionalized ILs, including [N_1888_][Ac], [N_1888_][FA], [N_1888_][Gly], and [P_66614_][Gly], was investigated under a gas concentration of 6.1% and an atmospheric pressure of 313.15 K, as shown in Figure 2a. The experimental results followed the order of [P_66614_][Gly] (130.62 mg/g) > [N_1888_][Gly] (101.42 mg/g) > [N_1888_][FA] (89.69 mg/g) > [N_1888_][Ac] (86.49 mg/g). In order to evaluate the absorption performance of DCM by the ILs designed in this work, we compared two ILs [Emim][Ac] [14] and [Beim][EtSO_4_] [24] that have been reported to have better absorption of DCM. The absorption capacity of [Emim][Ac] and [Beim][EtSO_4_] was tested under the same conditions in this study, as shown in Figure 2b. Among the four ILs, it is noteworthy that the absorption capacity of [P_66614_][Gly] was more than two times that of [Beim][EtSO_4_] and [Emim][Ac].

For the strong hydrophobic cations [N_1888_]^+^ and [P_66614_]^+^, the absorption capacity of [P_66614_][Gly] not only significantly improved the absorption capacity as compared to [N_1888_][Gly] but also reduced the time to reach absorption saturation. Nonpolar affinity [25] dominates the interaction between the cations and DCM, which may be the result of the higher nonpolar affinity between [P_66614_]^+^-DCM than [N_1888_]^+^-DCM. In addition, it is possible that increasing the alkyl chain length of the cations increased the steric hindrance between the ions, making it easier for the DCM molecules to approach the anions. Simultaneously, comparing [N_1888_][Ac] with [Emim][Ac], due to the nonpolar affinity of DCM with noncyclic cations, it exhibited higher absorption capacity for DCM than aromatic cations. [N_1888_][FA] had better absorption performance than [N_1888_][Ac], indicating that the length of the anionic alkyl chain was not conducive to DCM absorption, which verified the simulation calculation results of Wu et al. [25]. Under the same experimental conditions, it was clear from the results that among the three anions studied, [N_1888_][Gly] had the greatest DCM absorption capacity. The reason may be that [Gly]^−^ contains both amino and carboxyl groups, providing more active sites than [FA]^−^ and [Ac]^−^ [36]. These experimental results were further verified by quantum chemistry calculations.

#### 3.1.2. Effect of Temperature and Gas Concentration of DCM on the Absorption Capacity

The absorption experiments of [P_66614_][Gly] at different temperatures were conducted under the condition of 6.1% DCM concentration. As shown in Figure 3a, the absorption capacity of [P_66614_][Gly] decreased obviously as the temperature increased, which is also the common trend of most gas absorption; that is, it had higher solubility at low temperatures. In detail, the absorption capacity decreased from 130.62 to 60.29 mg/g, and the time for DCM to reach absorption saturation was shortened from 6 to 2 h as the temperature increased from 313.15 to 353.15 K. The results demonstrated that the absorption process was significantly impacted by temperature. Theoretically, the viscosity of [P_66614_][Gly] decreased when the temperature increased (Appendix A), the mass transfer rate accelerated [37], and the time of DCM to reach absorption saturation was shortened.

Figure 3b displays the change in the absorption capacity of [P_66614_][Gly] as a function of DCM gas concentration, and the gas concentrations were 4.2%, 6.1%, and 8.1%. At the absorption temperature of 313.15 K, when the inlet concentration increased from 4.2% to 8.1%, the DCM absorption capacity of [P_66614_][Gly] increased from 100.71 to 144 mg/g. Actually, in the absorption process with a specific absorption temperature, the partial pressure of DCM gas on the side of the gas film will increase when the inlet vapor concentration increases, and increasing the concentration gradient between gas and liquid phases boosts the mass transfer force [12]. This is conducive to the absorption of DCM in the ILs. According to the two-film theory [38], the higher gas concentration provides a higher force during the gas–liquid absorption process, and this effect facilitates the absorption of DCM in [P_66614_][Gly].

#### 3.1.3. Absorption-Desorption Cycles of [P_66614_][Gly]

In the practical application of absorbents, the regeneration performance is directly related to the operation cost and equipment investment.

According to previous reports in the literature, a higher temperature is beneficial to gas desorption from ILs [39]. In this study, since there was only DCM, the absorbed ILs were regenerated by passing nitrogen. The saturated absorbent was blown off with 100 mL/min N_2_ [40] for 10 h at 373.15 K under laboratory conditions. Different methods were used to obtain the mass change in the absorbent before and after desorption. When the mass of the absorbent before and after desorption was almost the same, the absorbent was considered to be desorbed completely. It can be seen from Figure 4a, the absorption capacity had no apparent loss after five cycles of absorption/desorption on [P_66614_][Gly], confirming that it had outstanding thermostability and could be reused. The FT-IR spectra of the regenerated [P_66614_][Gly] and fresh [P_66614_][Gly] are shown in Figure 4b. The FT-IR spectra of the regenerated [P_66614_][Gly] was consistent with the fresh IL, which proved that its structure did not change after recycling.

### 3.2. VLE of ILs and DCM

#### 3.2.1. Reliability of Experimental Equipment

First of all, a check of the experimental device was carried out. Then the VLE experiment of DCM (1) + [Emim][Ac] (2) (x_1_ = 0.9) was carried out, and the experimental values were compared with the data reported in the literature [14]. The comparison of vapor pressure values is shown in Appendix A, and it can be intuitively seen from the Figure that there is no significant deviation between the experimental values and the literature data, indicating that the device can be used to accurately measure the VLE data of the DCM + IL mixed solutions.

#### 3.2.2. Vapor Pressure Data of DCM + ILs Binary System

The vapor pressure of four binary systems DCM + [N_1888_][Ac], DCM + [N_1888_][FA], DCM + [N_1888_][Gly] and DCM + [P_66614_][Gly] at different DCM molar fractions and temperatures were tested as illustrated in Figure 5. An increase in the value of the vapor pressure accompanies the rise in temperature. The larger the molar fraction of IL, the smaller the vapor pressure values at the same temperature. At the same DCM molar fraction and temperature, the vapor pressures of the binary systems are as follows: DCM + [P_66614_][Gly] < DCM + [N_1888_][Gly] < DCM + [N_1888_][FA] < DCM + [N_1888_][Ac], further indicating that [P_66614_][Gly] has the tremendous potential for DCM absorption. The VLE data of the DCM + IL binary system were predicted on the basis of the NRTL model (Appendix A). Table 1 summarizes the fitted parameters; the average relative deviation (ARD) [41,42] and rRMSD [43] are defined as Equations (3) and (4), respectively.
(3)$$\mathrm{ARD}(P)=\left(\frac{1}{n}\sum_{i=1}^{n}\left|p^{\mathrm{NRTL}}-p^{\exp}\right|{/p}^{\exp}\right)$$
(4)$$\mathrm{rRMSD}(P)=\sqrt{\frac{1}{n}\sum_{i=1}^{n}{\left(\frac{p^{\exp}-p^{\mathrm{NRTL}}}{p^{\exp}}\right)}^{2}}$$

The experimental vapor pressure data and the data predicted by the NRTL model were summarized in (Appendix A) with an ARD of less than 2% and an overall rRMSD of 0.8467. These results indicate that the NRTL model was suitable for the DCM + IL binary system in this study, and provides an excellent thermodynamic method for designing and simulating the DCM absorption process.

### 3.3. Mechanism of DCM Absorption

#### 3.3.1. FT-IR and ^1^H-NMR Analysis

The interactions between the DCM and the [P_66614_][Gly] were studied by the FT-IR spectra and ^1^H-NMR. The FT-IR spectra of the DCM, fresh [P_66614_][Gly] and [P_66614_][Gly] after absorbing DCM, were recorded and are displayed in Figure 6a. It can be seen that a new absorption peak at 756 cm^−1^ was observed in the FT-IR spectra after the DCM absorption compared with the fresh [P_66614_][Gly], which was the C-Cl stretching vibration peak in DCM. The infrared spectrum also showed that the stretching vibration peak of C=O in the [Gly]^−^ shifted from 1574 cm^−1^ to 1595 cm^−1^, indicating that there was a C-H∙∙∙O hydrogen bonding interaction [44,45] between the [Gly]^−^ and the DCM. At the same time, there may be a C-H∙∙∙Cl hydrogen bonding interaction [24], as the bending vibration peak of -CH_2_ in the [P_66614_]^+^ at 1305 cm^−1^ moved to 1309 cm^−1^.

To further understand the absorption mechanism, the ILs before and after the absorption of the DCM were characterized by ^1^H-NMR spectra. From Figure 6b, it can be seen that the peak corresponding to the DCM appeared at 5.76 ppm, and there was no evident chemical shift in the ^1^H-NMR spectra of [P_66614_][Gly] before and after the DCM absorption, which means that no new substance was formed after the [P_66614_][Gly] absorption of DCM.

#### 3.3.2. Model and Calculation Section

##### Effect of Anion and Cation Structures on DCM Absorption

The affinity strength between the molecules was analyzed utilizing the σ-profile obtained from the COSMO model in this work. Since the [Gly]^−^, [N_1888_]^+^, and [P_66614_]^+^ are not included in the built-in database of COSMOthermX Version 19.0.4 [14], the optimized anions were imported into the COSMObase through Gaussian 09, revision D.01. Figure 7 shows the σ-profiles of [Ac]^−^, [FA]^−^, [Gly]^−^, [N_1888_]^+^, [P_66614_]^+^, and DCM. The σ-profile consists of three distinct regions: the hydrogen bond donor (HBD) region (σ < −0.0084 e/Å^2^), the nonpolar region (−0.0084 e/Å^2^ < σ < 0.0084 e/Å^2^), and the hydrogen bond acceptor (HBA) region (σ > 0.0084 e/Å^2^) [46].

Most peaks of the σ-profile are in the nonpolar region for DCM, and it has a high attraction for other nonpolar components. A faint peak can be seen in the HBD region in addition to the nonpolar zone, suggesting that the hydrogen bonds may be formed between the DCM and the other HBA. It can be seen from Figure 7 that the nonpolar area is the main distribution of the σ-profile of [N_1888_]^+^ and [P_66614_]^+^, suggesting that the interaction is mainly a nonpolar affinity between the cations and the DCM. It also can be seen that the peak of [P_66614_]^+^ was higher than that of [N_1888_]^+^, indicating that the nonpolar affinity between [P_66614_]^+^-DCM is higher than that of [N_1888_]^+^-DCM. For the σ-profile of anions [Ac]^−^, [FA]^−^, and [Gly]^−^, they are mainly in the HBA region and the nonpolar region, while the strong peak of the σ-profile appears around the 0.02 e/Å^2^ HBA region, indicating that the interactions between the anions and the DCM are mainly hydrogen bonds.

The calculated results of the interaction energy of the anion and cation of the ILs with the DCM are shown in Figure 8, and the optimized structures are presented in Appendix A. The interaction energies of [N_1888_]^+^-DCM and [P_66614_]^+^-DCM were −37.26 kJ∙mol^−1^ and −37.73 kJ∙mol^−1^, respectively, while the interaction energies of [Ac]^−^-DCM, [FA]^−^-DCM, and [Gly]^−^-DCM were −78.67 kJ∙mol^−1^, 76.64 kJ∙mol^−1^, and −74.81 kJ∙mol^−1^, respectively. The distances of O∙∙∙H were 2.10 Å, 2.25 Å, 2.19 Å, 2.19 Å, 2.26 Å, and 2.12 Å. These bond distances were greater than the covalent bond length of O-H (0.96 Å) and shorter than the Van der Waals distance between atom O and atom H (2.72 Å), which was within acceptable standards for a hydrogen bond [47]. The distances of Cl∙∙∙H were around 2.9 Å for cation-DCM systems, which also falls into the range 2.7 Å–3.0 Å of C-H∙∙∙Cl hydrogen bond [48], suggesting a weak interaction between the cation and the DCM. In general, the interaction energy of the anion–DCM was more negative than that of the cation–DCM, which indicates that the anion played a crucial role in the absorption of the DCM.

##### Interaction Energy Analysis

The optimized structures of the [P_66614_][Gly]-DCM, [N_1888_][Gly]-DCM, [N_1888_][FA]-DCM, and [N_1888_][Ac]-DCM are illustrated in Figure 9. Table 2 lists the interaction energies of the IL-DCM as well as other information. The bond angle ∠X-H∙∙∙Y was larger than 90° [49], and the hydrogen bonds were within the accepted norm. There was a C-H∙∙∙O hydrogen bond interaction in [P_66614_][Gly], and the bond distance and bond angle was 1.91 Å and 171.15°, respectively. In addition to the hydrogen bond with [Gly]^−^ in the anion, a weaker hydrogen bond also formed between the DCM and [P_66614_]^+^, which was C-H∙∙∙Cl (2.89 Å). As shown in Table 2, the interaction energy of the IL-DCM follows the order of [P_66614_][Gly]-DCM(−78.28 kJ/mol) > [N_1888_][Gly]-DCM(−64.24 kJ/mol) > [N_1888_][FA]-DCM(−58.58 kJ/mol) > [N_1888_][Ac]-DCM(−55.37 kJ/mol). The magnitude of the interaction energy indicates the strength of the interaction between the DCM and the absorbents. The intensity of the interaction between the absorbents and the DCM was positively correlated with the absorption capacity. In general, [P_66614_][Gly]-DCM has the largest interaction energy among the four complexes, which explains the results of the absorption experiments.

## 4. Conclusions

In this study, four carboxyl-functionalized ILs were synthesized, and their absorption capacity were measured by a set of absorption experiments. The results showed that the [P_66614_][Gly] exhibited a higher absorption capacity of 130.62 mg DCM/g IL at 313.15 K and 6.1% DCM concentration than the reported ILs [Emim][Ac] and [Beim][EtSO_4_]. Subsequently, the vapor pressure data of the DCM + IL binary systems were measured by a VLE experimental device and predicated by the NRTL model. According to the fitted results, the model worked adequately for the VLE of these binary systems. The order of the vapor pressure of the DCM + IL binary system under the same conditions was DCM + [P_66614_][Gly] < DCM + [N_1888_][Gly] < DCM + [N_1888_][FA] < DCM + [N_1888_][Ac]. In addition, the [P_66614_][Gly] exhibited good recyclability, with the absorption capacity showing no significant decrease after five cycles of absorption–desorption. The absorption mechanism was identified through FT-IR, ^1^H-NMR and quantum chemistry calculations, indicating that the hydrogen bond played a significant role in the absorption of DCM by [P_66614_][Gly]. In conclusion, [P_66614_][Gly] was found to be a promising, efficient, and green absorbent for DCM.

## Figures and Tables

**Figure 1 ijerph-20-05787-f001:**
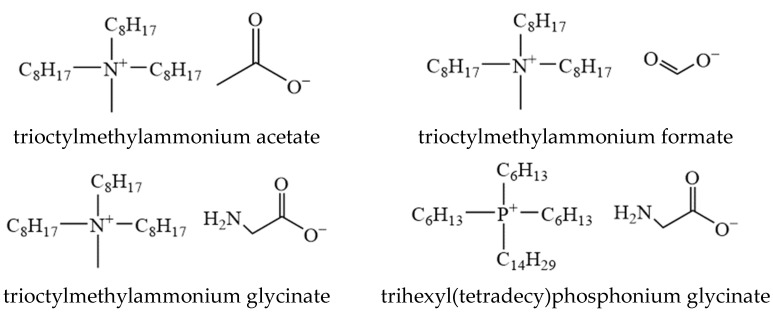
The structure of ILs used in this work.

**Figure 2 ijerph-20-05787-f002:**
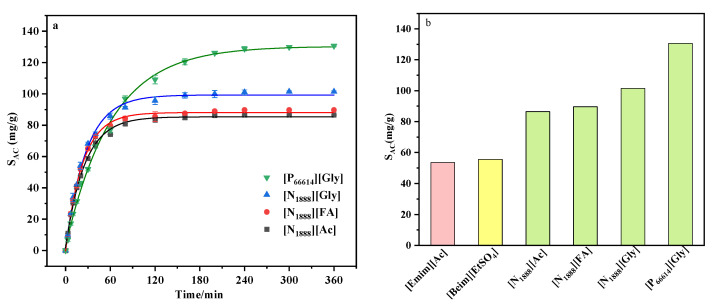
Absorption capacity (**a**) of different ILs at 313.15 K and 6.1% concentration and comparison (**b**) of ILs synthesized in this study with have been reported ILs. (Green represents the DCM absorption capacity of the ILs prepared in this study, and pink and yellow represent the DCM absorption capacity of reported ILs [Emim] [Ac] and [Beim] [EtSO_4_].)

**Figure 3 ijerph-20-05787-f003:**
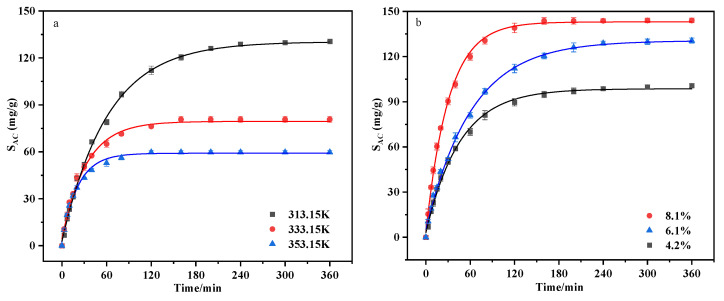
Absorption capacity of [P_66614_][Gly] for DCM at different temperatures (**a**) with DCM concentration of 6.1% and different DCM concentrations (**b**) at 313.15 K.

**Figure 4 ijerph-20-05787-f004:**
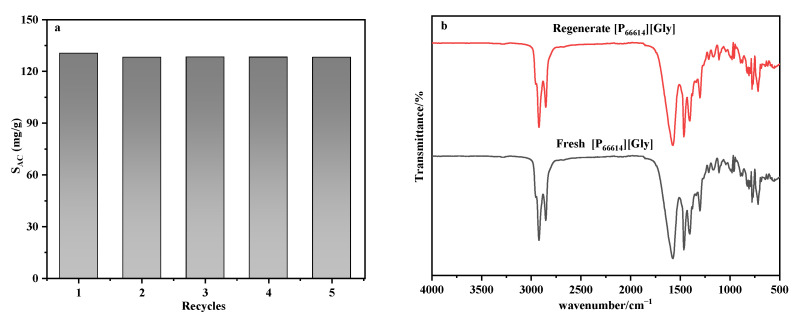
DCM absorption capacity of the recycled [P_66614_][Gly] at 313.15 K (**a**) and FT-IR spectra (**b**) of fresh and regenerated [P_66614_][Gly].

**Figure 5 ijerph-20-05787-f005:**
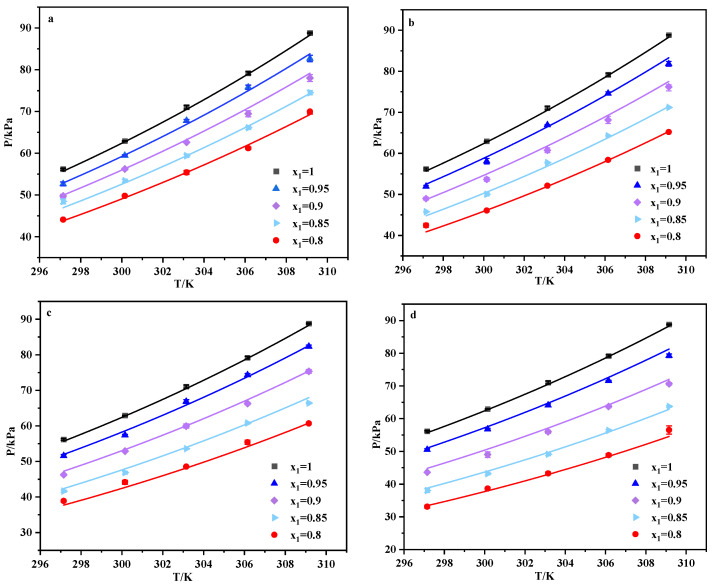
VLE of binary of DCM (1) + [N_1888_][Ac] (2) (**a**), DCM (1) + [N_1888_][FA] (2) (**b**), DCM (1) + [N_1888_][Gly] (2) (**c**), DCM (1) + [P_66614_][Gly] (2) (**d**) at different temperatures. (x_1_ denotes the mole fraction of DCM; scattered points, experimental values; solid lines, values predicted by the NRTL model).

**Figure 6 ijerph-20-05787-f006:**
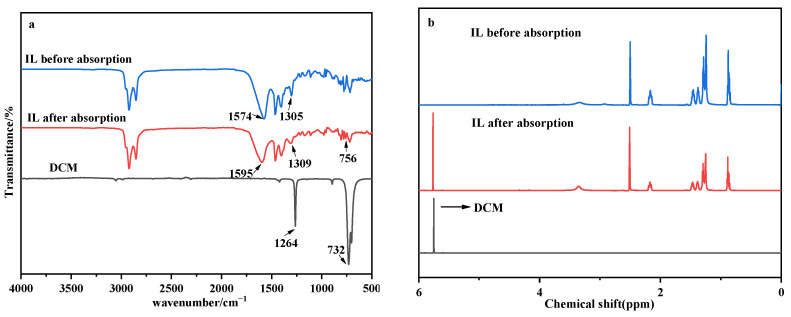
FT-IR (**a**) spectra and ^1^H-NMR (**b**) of [P_66614_][Gly] before and after absorption of DCM.

**Figure 7 ijerph-20-05787-f007:**
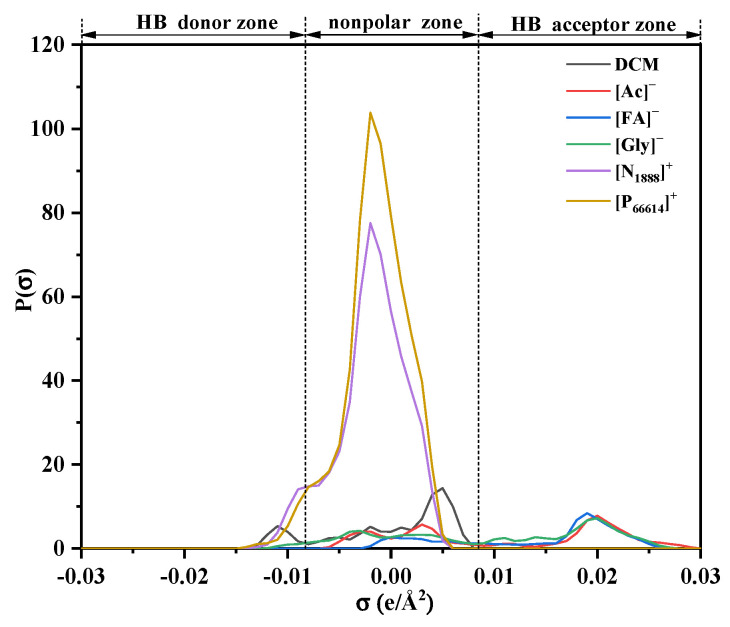
σ-profiles of DCM, [Ac]^−^, [FA]^−^, [Gly]^−^, [N_1888_]^+^, and [P_66614_]^+^.

**Figure 8 ijerph-20-05787-f008:**
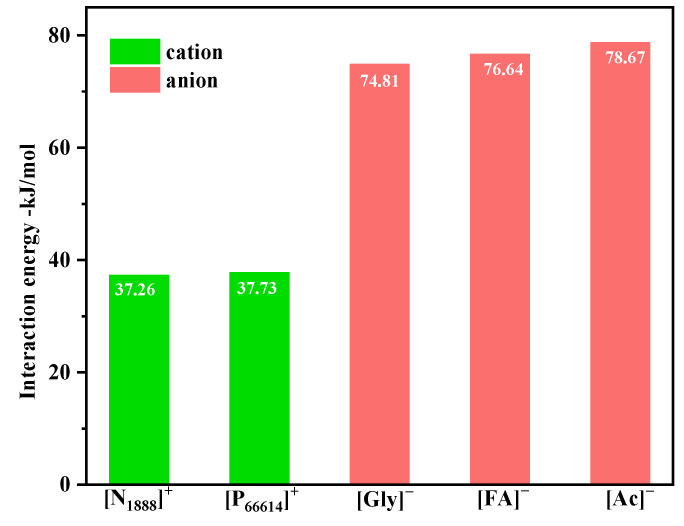
Calculated results of interaction energy of anion and cation of ILs with DCM.

**Figure 9 ijerph-20-05787-f009:**
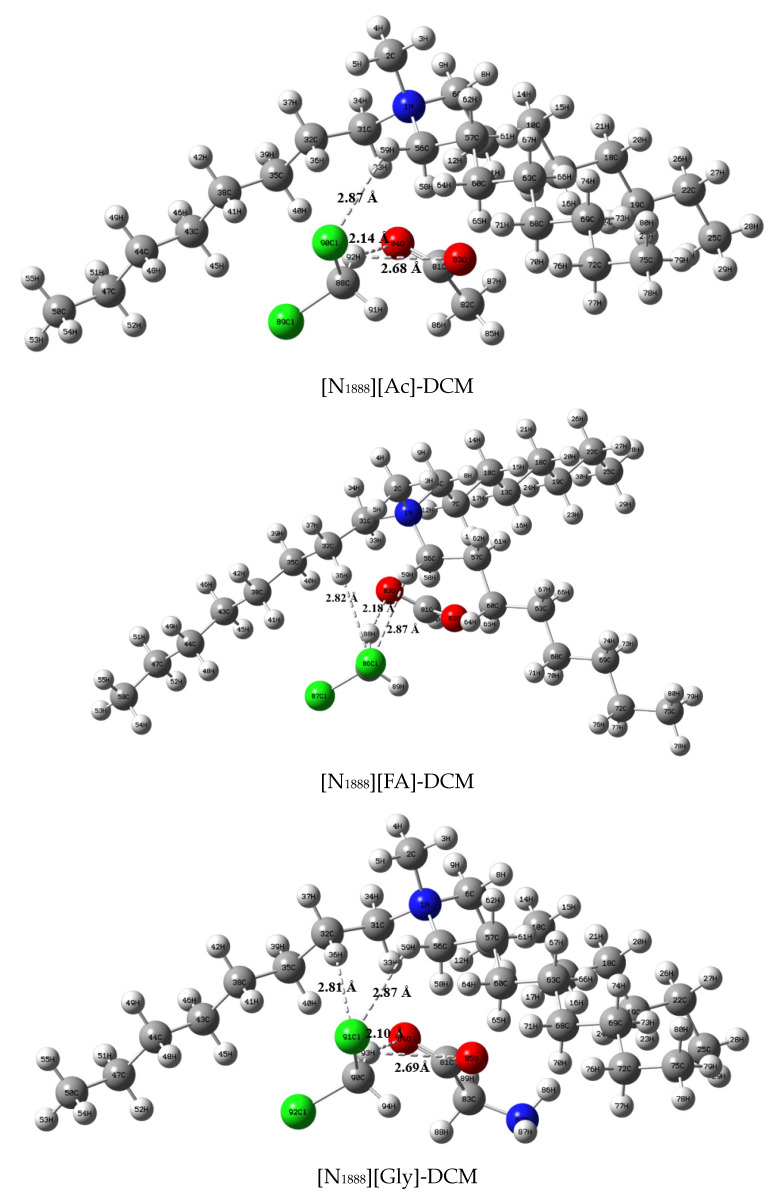
Optimized structures of the ILs–DCM complex.

**Table 1 ijerph-20-05787-t001:** The binary NRTL parameters fitted.

Binary System	α_12_	g_12_–g_22_	g_21_–g_11_	ARD	rRMSD
		J·mol^−1^	J·mol^−1^		
DCM + [N_1888_][Ac]	0.0686	−23,396.2	13,497.93	0.0081	0.6620
DCM + [N_1888_][FA]	0.9900	210.1503	−264.229	0.0105	0.7840
DCM + [N_1888_][Gly]	0.9900	274.86	−342.058	0.0121	0.8150
DCM + [P_66614_][Gly]	0.9900	358.2333	−422.142	0.0152	1.0900

**Table 2 ijerph-20-05787-t002:** Bond distance, bond angle, and interaction energy of ILs–DCM.

IL-DCM	Atomic Number	Bond Distance (Å)	Bond Angle (deg)	Δ*E*/(kJ/mol)
[N_1888_][Ac]-DCM	88C-92H···84O	2.14	149.39	−55.37
	88C-92H···83O	2.68	107.17	
	56C-59H∙∙∙90Cl	2.87	128.22	
[N_1888_][FA]-DCM	85C-88H···83O	2.18	147.53	−58.58
	32C-36H···86Cl	2.82	162.71	
	56C-59H···86Cl	2.87	129.49	
[N_1888_][Gly]-DCM	90C-93H···84O	2.10	152.83	−62.24
	90C-93H···85O	2.69	109.85	
	32C-36H···91Cl	2.81	161.45	
	56C-59H···91Cl	2.87	128.35	
[P_66614_][Gly]-DCM	111C-115H···106O	1.91	171.15	−78.28
	3C-36H···113Cl	2.89	162.93	

## Data Availability

The authors declare that all the data and materials are available to be shared upon formal request.

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
