# Peer review of "Effective Absorption of Dichloromethane Using Carboxyl-Functionalized Ionic Liquids"

_ijerph, 2023, doi:10.3390/ijerph20105787_

Round 1
Reviewer 1 Report
Dear Authors,
It is found that authors worked very hard from your manuscript.
I was wondering if you could add or revise sentence based on comments as below;
Pages 3, Lines on 115
What kinds of the generated byproduct was removed?
What is a reason for a removal of the generated byproduct?
Pages 3, Lines on 136-137
“the water content of these four ILs was lower than 2000 ppm”
→ What ppm value of the water content is generally applicable?
Pages 4, Lines on 174-175
“observe whether the two sides of the U- shaped tube could remain at the same level for about 20 minutes.”
→ How did you observe the level?
Pages 11, Lines on 393-394
“the five cycles of adsorption-desorption”
→ How many cycles are assumed as the industrial use?
Sincerely.
Author Response
Dear Reviewer,
Thank you very much for considering our manuscript (ijerph-2269389) entitled “Effective absorption of dichloromethane using carboxyl-functionalized ionic liquids”.
We have carefully considered all the comments and prepared item-by-item responses to each comment. All comments were considered and the appropriate changes are explained one by one in the response. We are sincerely grateful to the reviewers for their fruitful comments, which have led to significant improvements in the quality and clarity of this manuscript.
Thank you very much for all your help and looking forward to hearing from you soon.
Sincerely
Xiangping Zhang
Institute of Process Engineering
Chinese Academy of Sciences
- O. Box 353
Beijing 100190
- R. China

Reviewer 2 Report
The manuscript reported the adsorption of four carboxyl functionalized Ils, including trioctylmethylammonium acetate [N1888] [Ac], trioctylmethylammonium formate [N1888] [FA], trioctylmethylammonium glycinate [N1888] [Gly], and trihexyl(tetradecyl)phosphonium glycinate [P66614] [Gly]. The authors reported the various factors that affected the adsorption capacity and the interaction mechanism. The manuscript shows a good IL adsorbent for the removal of DCM.
Please see the following comments before publication:
A- Abstract section
1- It is better to state the actual work and challenges in the first sentence.
2- It is highly recommended to write the full name for the first time and then use the abbreviation.
B- The results and discussion
1- In line 216, the citation of Wu et al is missing. Please revise and add the corresponding citation.
2- It is better to redesign the X and Y axis labels to be more clear and more readable.
3- It is highly recommended to revise this sentence to be more clear and readable “Under the same experimental conditions, it was clear from the results that among the three kinds of anions studied, [Gly]- had the greatest DCM absorption capacity, which is due to carboxylate- based anions having more polar peak[25], and [FA]- and [Ac]- showed comparable DCM absorption capacity, which may be due to [Gly]- providing more active sites than [FA]- and [Ac]-, which contain both amino and carboxyl groups[36]”.
4- It is highly recommended to revise the manuscript`s language.
5- It is a very complicated sentence “It can be seen from Figure 7. that the nonpolar area is where the σ-profile of [N1888]+ and [P66614]+ are primarily distributed, suggesting that the interaction is mainly nonpolar affinity between cations and DCM, and found the peak of [P66614]+ was higher than that of [N1888]+, which indicates that the nonpolar affinity between [P66614]+-DCM is higher than that of [N1888]+-DCM. ”. It is highly recommended to revise and redesign this sentence.
Reviewer 3 Report
The article „Effective absorption of dichloromethane using carboxyl-functionalized ionic liquids“ from authors Mengjun Wang, Manman Zhang, Shaojuan Zeng, Yi Nie, Tao Li, Baozeng Ren, Yinge Bai and Xiangping Zhang describes research results dealing with absorption of dichloromethane vapours using four synthetized hydrophobic ionic liquids. The effect of both IL cation and anion structures is discussed and explained in this article utilizing IR and NMR spectroscopies and quantum chemistry calculations.
This article serves as very interesting for readers of International Journal of Environmental Research and Public Health.
However, the submitted article still needs revisions:
1. Page 1, Abstract, lines 19-23: Please, add the Reference(s) number(s) to this claims or redefine the sentence.
2. Page 1, Abstract, line 23: Please, explain the abbreviation NRTL.
3. Page 2, line 47: Please, correct the DEHP and DEHA titles: “di(2-ethylhexyl) phthalate (DEHP), bis(2-ethylhexyl) adipate (DEHA)“
4. Page 2, line 63: Please, (re)consider the claims dealing with absorption capacity of Beim.EtSO4 = 0.426 g/g (versus 55.56 mg/g mentioned in line 202, page 5).
5. Page 2, lines 76-77: The sentence: “Overall, it is… “ seems to be confusing.
6. Page 2, line 79: please, reconsider: “…such as quaternary and quaternary phosphine…” Did you mean “quaternary ammonium and quaternary phosphonium”?
7. Table S1 and Figure S9: Used units are confusing. Did you mean millipascal.second (mPa.s)? The same line 177 (kilopascal = kPa).
8. Page 5, line 202: ” …have been reported” Please, add the reference(s) number(s).
9. Page 5, line 209-210: “…is higher that of N1888-DCM” Please, add the Reference(s) or verification of this claims.
10. Page 5, line 216: Please, add the Reference number.
11. Page 9, line 332: Please, consider the mentioned HBA region.

Author Response

(The authors gave the same response as above.)

Reviewer 4 Report
1. There are a lot of ILs. It is wondered why the authors selected the following type ILs, [N1888][Ac], [N1888][FA], [N1888][Gly], and [P66614][Gly]. What’s the importance of KH to this work when making the selection?
2. Please provide equipment schematic for absorption.
3. Please summarize the absorption capacity of ILs in previous research and make a comparison to indicate the position of the ILs in current work.
4. The structures and names in Figure 1 and Figure 9 are not clear.
5. The authors indicated that the absorption capacity variation was mainly due to interaction energy difference. For ILs, another factor might be involved is the crystallinity.
Round 2
Reviewer 2 Report
The authors have raised most of the comments. So, I recommend accepting the manuscript in its current form.
Author Response
Dear Reviewer,
Thank you very much for considering our manuscript (ijerph-2269389) entitled “Effective absorption of dichloromethane using carboxyl-functionalized ionic liquids”.
Reviewer 4 Report
Accept
Author Response

(The authors gave the same response as above.)
